# Autophagy and Female Fertility: Mechanisms, Clinical Implications, and Emerging Therapies

**DOI:** 10.3390/cells13161354

**Published:** 2024-08-14

**Authors:** Abdel Halim Harrath, Md Ataur Rahman, Sujay Kumar Bhajan, Anup Kumar Bishwas, MD. Hasanur Rahman, Saleh Alwasel, Maroua Jalouli, Sojin Kang, Moon Nyeo Park, Bonglee Kim

**Affiliations:** 1Zoology Department, College of Science, King Saud University, Riyadh 11451, Saudi Arabia; hharrath@ksu.edu.sa (A.H.H.); salwasel@ksu.edu.sa (S.A.); 2Department of Neurology, University of Michigan, Ann Arbor, MI 48109, USA; ataur1981rahman@hotmail.com; 3Department of Biotechnology and Genetic Engineering, Faculty of Life Sciences, Bangabandhu Sheikh Mujibur Rahman Science and Technology University, Gopalganj 8100, Bangladesh; sujaybge@gmail.com (S.K.B.); anupkumarbishwas21@gmail.com (A.K.B.); hasanurrahman.bge@gmail.com (M.H.R.); 4Department of Biology, College of Science, Imam Mohammad Ibn Saud Islamic University (IMSIU), Riyadh 11623, Saudi Arabia; mejalouli@imamu.edu.sa; 5Department of Pathology, College of Korean Medicine, Kyung Hee University, 1-5 Hoegidong Dongdaemun-gu, Seoul 02447, Republic of Korea; bmb1994@khu.ac.kr (S.K.); mnpark@khu.ac.kr (M.N.P.); 6Korean Medicine-Based Drug Repositioning Cancer Research Center, College of Korean Medicine, Kyung Hee University, Seoul 02447, Republic of Korea

**Keywords:** autophagy, female fertility, oocyte quality, follicular development, reproductive aging, therapeutic interventions

## Abstract

Autophagy, an evolutionarily conserved cellular mechanism essential for maintaining internal stability, plays a crucial function in female reproductive ability. In this review, we discuss the complex interplay between autophagy and several facets of female reproductive health, encompassing pregnancy, ovarian functions, gynecologic malignancies, endometriosis, and infertility. Existing research emphasizes the crucial significance of autophagy in embryo implantation, specifically in the endometrium, highlighting its necessity in ensuring proper fetal development. Although some knowledge has been gained, there is still a lack of research on the specific molecular impacts of autophagy on the quality of oocytes, the growth of follicles, and general reproductive health. Autophagy plays a role in the maturation, quality, and development of oocytes. It is also involved in reproductive aging, contributing to reductions in reproductive function that occur with age. This review explores the physiological functions of autophagy in the female reproductive system, its participation in reproductive toxicity, and its important connections with the endometrium and embryo. In addition, this study investigates the possibility of emerging treatment approaches that aim to modify autophagy, using both natural substances and synthetic molecules, to improve female fertility and reproductive outcomes. Additionally, this review intends to inspire future exploration into the intricate role of autophagy in female reproductive health by reviewing recent studies and pinpointing areas where current knowledge is lacking. Subsequent investigations should prioritize the conversion of these discoveries into practical uses in the medical field, which could potentially result in groundbreaking therapies for infertility and other difficulties related to reproduction. Therefore, gaining a comprehensive understanding of the many effects of autophagy on female fertility would not only further the field of reproductive biology but also open new possibilities for diagnostic and treatment methods.

## 1. Introduction

Autophagy is crucial to cellular homeostasis in the complex network of cellular biology and controls self-degradation and recycling of cellular components that are essential to cell health [1]. Although its significance in cellular homeostasis is well established, recent investigations have demonstrated its complex participation in physiological and pathological processes, including female reproductive health [2]. Autophagy-related genes (ATGs) lead to complicated autophagic machinery. Beclin-1, ATG5, and ATG7 are key elements of this ensemble, whereby they guide autophagosome formation and cellular cargo breakdown [3]. Recently, it has become well known that autophagy is crucial for understanding its complex effects on female reproductive health [4]. With a delicately coordinated network of organs and hormonal signals, the female reproductive system depends on cellular equilibrium [5]. Recent investigations have linked autophagy to oocyte maturation, embryonic development, and placental function. Infertility and pregnancy difficulties have been linked to autophagy dysregulation [6]. Autophagy has far-reaching effects beyond cellular maintenance, and the complex role of autophagy in female reproductive health has not yet been well studied.

The aim of this study was to explore the functional aspects of autophagy in the female reproductive system beyond molecular details. This finding may explain the complex interaction of ATG proteins and signaling networks that regulate autophagy. An understanding of these molecular details is crucial for elucidating the reproductive health effects of autophagy [7]. Autophagy protects reproductive systems and processes, ranging from oocyte quality control to early embryonic development [8]. Autophagy also affects neurological illnesses, carcinogenesis, diabetes, development, life expectancy, and fertility [9]. During the process of follicular atresia, the remaining follicles undergo a process of degeneration and finally disintegration. The quantity of remaining oocytes, as well as their quality, continues to decrease with increasing age, which is the cause of the age-related decrease in female fertility [10]. There is a correlation between the quality of the oocyte and the development of the embryo and the fetus, as evidenced by the fact that older women have a greater chance of miscarriage [11]. To investigate the effects of autophagic dysregulation on polycystic ovary syndrome (PCOS), endometriosis, and recurrent pregnancy loss, in this review, we explored autophagy and its effects on female reproductive health in depth.

## 2. Crucial Relationship between Autophagy and Follicular Development

Follicular development is a complex process that is crucial for the fertility and reproduction of females. The follicles within the ovaries progress through a sequence of stages, thus culminating in ovulation [12]. Autophagy has a notable impact on multiple physiological processes, such as follicular growth [13]. An understanding of the complex interplay between autophagy and follicular development is crucial for comprehending ovarian function and fertility. Autophagy has multiple functions at various stages of folliculogenesis in ovarian follicles, ranging from the primordial to the preovulatory stages [14,15,16,17]. Primordial follicles, which are inactive follicles, depend on autophagy to maintain their dormancy and ensure their survival [18]. Autophagy inhibits the early activation of primordial follicles, hence controlling both the quantity and quality of the follicular pool [19]. During the shift from the dormant stage to the developing stage, autophagy becomes crucial for the functioning of granulosa cells and the quality of oocytes [19,20]. The granulosa cells that surround the oocyte play a vital role in supporting its development [21]. Autophagy in granulosa cells is essential for the appropriate turnover of organelles, the generation of energy, and the secretion of hormones, which are all crucial for the growth and maturation of follicles [22]. Moreover, autophagy protects against damage caused by oxidative stress throughout the process of follicular growth [23]. Oxidative stress caused by reactive oxygen species (ROS) produced in the follicle can negatively affect the quality of oocytes and reduce fertility [24]. Autophagy facilitates the reduction of oxidative stress by eliminating impaired cellular elements, thereby protecting the health of follicles and enhancing the competence of oocytes [25]. However, the regulation of autophagy in the ovaries is rigorously governed by many signaling pathways and molecular mechanisms. Autophagy is controlled by important regulators such as mTOR and AMPK [26,27]. These regulators use environmental cues and hormone signals to adjust autophagic activity based on the availability of nutrients, energy levels, and stressful conditions. The control of autophagy in the ovary is crucial, especially during the formation of follicles [28]. Gonadotropins, which include follicle-stimulating hormone (FSH) and luteinizing hormone (LH), control the process of autophagy in granulosa cells and theca cells, which correspondingly affects the formation of follicles and the production of steroids [19]. Moreover, ovarian hormones such as estrogen and progesterone have distinct impacts on autophagy in a specific manner according to the stage, thus emphasizing the complex interaction between autophagy and ovarian physiology [29].

Autophagy has diverse functions during folliculogenesis, including the regulation of primordial follicle quiescence and the support of granulosa cell function and oocyte quality [24]. An understanding of the control and abnormal function of autophagy in the ovary will provide valuable insights into the underlying causes of ovarian diseases and possible approaches for treating reproductive diseases [30]. Moreover, the recognition of the essential connection between autophagy and follicular growth has important practical implications for the fields of reproductive medicine, as well as fertility preservation [31]. Impaired autophagy has been associated with ovarian disorders, including premature ovarian insufficiency (POI) and polycystic ovary syndrome (PCOS), thus highlighting the significance of regulating autophagy as a potential treatment approach [32] (Figure 1). Future studies should prioritize investigating the molecular processes that govern the control of autophagy in the ovary and its influence on follicular development. Furthermore, the exploration of the therapeutic possibilities of directing autophagic pathways toward the treatment of ovarian diseases and infertility has potential for enhancing reproductive outcomes and the well-being of women.

## 3. Emerging Role of Autophagy in Oocyte Quality Control

Egg cells, which are also known as oocytes, are essential for reproduction and have a significant influence on both embryonic development and fertility [33]. It is essential to understand the complex processes regulating oocyte quality (including the function of autophagy) to advance reproductive medicine and fertility treatments [34]. Autophagy in oocytes is involved in many phases of early embryonic growth, folliculogenesis, and oogenesis [19]. Oocyte quality has been shown to be significantly impacted by autophagy, which is a cellular process that is known for its ability to recycle and preserve cellular homeostasis. Oogenesis is a highly regulated process in which oocyte maturation occurs through complex molecular and cellular processes [35]. Autophagy plays a crucial role in various important processes of oocyte development, such as removing excessive or damaged organelles, such as mitochondria, which are necessary for maintaining oocyte quality [36]. Malfunctioning mitochondria can result in oxidative stress, as well as impaired oocyte quality, thus ultimately impacting the process of fertilization and the development of embryos [37]. Autophagy is a cellular process that selectively eliminates damaged mitochondria to maintain their integrity [38]. This process is known as mitophagy. 

Moreover, autophagy also impacts the survival and development of oocytes by controlling energy metabolism and nutrient availability [39]. Oocytes rely heavily on energy, thus necessitating adequate levels of ATP and metabolic substrates to undergo appropriate maturation and fertilization [40]. Autophagy is a process that breaks down and recycles cellular components to provide nutrients, which helps in the metabolism and development of oocytes [41]. Autophagy plays a significant role not only in the maturation of oocytes but also in the early stages of embryonic development [42]. After fertilization, the zygote undergoes a sequence of cellular divisions along with morphogenetic processes, thus resulting in the creation of a blastocyst [43]. Autophagy plays a role in embryonic development by facilitating the removal of paternal mitochondria, preserving the integrity of the genome, and promoting cellular differentiation [44]. Autophagy impairment during the initial stages of embryonic development can result in irregularities in development and growth, as well as difficulties during pregnancy [45].

The regulation of autophagy in oocytes is controlled by a complex system of signaling routes and molecular processes. Crucial controllers of autophagy, such as the mammalian target of rapamycin (mTOR) pathway and the AMP-activated protein kinase (AMPK) pathway, have essential functions in organizing autophagic activity in response to different environmental signals and cellular stresses [19,46]. Autophagy in oocytes is influenced by hormonal cues, growth factors, and nutrient availability, thus demonstrating the close relationship between metabolic conditions and reproductive function [47]. Ultimately, autophagy is crucial for controlling the quality of oocytes by participating in mitochondrial dynamics, energy metabolism, and cellular homeostasis [48]. An understanding of the complex relationship between autophagy and oocyte development has significant potential for developing reproductive medicine and fertility treatments, which can ultimately help people with infertility and reproductive diseases [49]. Although there have been notable advancements in comprehending the function of autophagy in oocyte quality (Figure 2), there are still several unresolved questions. Moreover, investigating prospective treatment strategies that focus on autophagy has potential for enhancing reproductive outcomes and addressing disorders associated with infertility.

Recent study has emphasized the crucial significance of autophagy in the process of oocyte maturation and follicle growth [50]. Investigations involving the deactivation of autophagic genes in oocytes have yielded valuable knowledge on the processes that determine the quality and developmental potential of oocytes [51]. For example, when important autophagic genes like Atg5 and Beclin1 are deactivated, it has been demonstrated that oocyte maturation is hindered, resulting in decreased fertility and worse embryonic development. Autophagy is a crucial process for eliminating impaired organelles and ensuring the stability of cellular conditions in oocytes. Nevertheless, despite these progressions, there is still a lack of literature that particularly focuses on the elimination of autophagic genes in oocytes. To close this gap, additional study is required to clarify the specific functions of several autophagic pathways and their interconnections in the process of oocyte development. These investigations will not only improve our understanding of reproductive biology but also potentially result in new therapeutic approaches for enhancing oocyte quality and fertility outcomes.

## 4. Role of Autophagy in Reproductive Aging

Reproductive aging is an inherent process marked by a gradual decrease in reproductive capacity as individuals grow older, and it impacts both males and females [52]. Although significant focus has been given to the hormonal and physiological transformations that occur throughout this period, recent studies have provided insight into the involvement of autophagy in reproductive aging [53]. A comprehensive understanding of the role of autophagy in the process of reproductive aging could provide valuable knowledge about the fundamental mechanisms that are involved and potential approaches for treatment [54]. The quality of oocytes is a crucial factor in determining fertility and reproductive success in females [55]. Oocytes are more susceptible to age-related deterioration because of the gradual increase in cellular damage. Research has demonstrated that the activity of autophagy declines in oocytes as they age, thus resulting in diminished quality and decreased ability to develop [56]. Impaired autophagy can lead to the buildup of impaired cell structures, such as mitochondria, and harmful clumps of proteins, which can contribute to the aging of egg cells and infertility [57]. The use of pharmacological or genetic methods to increase autophagic activity has been proposed as being a viable technique to enhance oocyte quality and fertility in elderly females [58]. The decrease in reproductive capacity due to aging is affected not only by internal factors but also by external factors such as lifestyle, nutrition, and environmental exposures [59]. Recent research indicates that dietary therapies and lifestyle changes that stimulate autophagy activation may have positive impacts on reproductive aging [60]. Studies have demonstrated that restricting caloric intake, practicing intermittent fasting, and supplementing the diet with chemicals that promote autophagy can increase autophagic activity and enhance reproductive outcomes in animal models [61,62]. Moreover, current research has demonstrated specific molecular pathways and regulatory systems that control autophagy during the process of reproductive aging (Figure 3). Autophagy is controlled by important regulators such as mTOR and AMPK. These regulators have been found to play a role in controlling the function of the ovaries and testes [63]. The exploration of the relationship between these communication routes and autophagy may result in the identification of new treatment targets for age-related loss of reproductive function [64]. 

mTOR serves as a pivotal controller of cell growth, proliferation, and metabolism. It combines signals from nutrients, growth factors, and cellular energy levels [65]. More precisely, mTOR signaling plays a vital role in initiating the growth of primordial follicles, an essential process in the formation of ovarian follicles [66]. Primordial follicle activation refers to the transformation of inactive primordial follicles into developing primary follicles [67]. This process is carefully controlled by an intricate network of signaling channels. mTOR facilitates this transition by augmenting protein synthesis, food absorption, and energy generation, thereby supplying the essential resources for follicle growth and development [68]. Furthermore, the mTOR signaling system interacts with the PI3K-Akt pathway to guarantee accurate follicle activation and growth. Imbalance in mTOR activity can result in conditions like premature ovarian failure or PCOS, underscoring its crucial involvement in preserving ovarian function and fertility [69].

Autophagy is crucial in reproductive aging because it helps to maintain cellular balance and protects reproductive function [64]. Impaired autophagy has been linked to the deterioration of oocyte and sperm quality, thus resulting in age-related infertility in both males and females [19]. Strategies focused on increasing autophagic activity show potential for maintaining reproductive health and fertility in elderly people [53]. Additional investigations are required to clarify the precise molecular pathways that underlie the involvement of autophagy in reproductive aging. Therefore, further research is necessary to identify specific therapies that can alleviate the reduction in fertility associated with aging.

## 5. Physiological Role of Autophagy in the Female Reproductive System

Autophagy has a complex impact on female reproductive physiology, whereby it affects several elements of fertility, pregnancy, and reproductive health [6]. An understanding of the complex relationship between autophagy and reproductive processes has the potential to identify the mechanisms that cause infertility, pregnancy problems, and reproductive disorders [7] (Figure 4). Furthermore, directing attention toward autophagic pathways may demonstrate innovative therapeutic approaches for the treatment of reproductive diseases and the enhancement of reproductive outcomes in women. Additional investigations are necessary to clarify the exact processes and therapeutic possibilities involved in manipulating autophagy in female reproductive disorders. Recently, scientists examined the role of autophagy in the complex processes of female reproductive physiology. This research has provided insights into how autophagy affects fertility, pregnancy, and reproductive diseases [7,70]. Autophagy plays a significant role in oogenesis, which is the process of egg production and a crucial aspect of female reproductive physiology [33]. Autophagy is essential for the growth and maintenance of oocytes, thus ensuring that they mature correctly and are capable of fertilization [71]. Research has shown that when the process of autophagy in oocytes is not properly regulated, it may result in a reduced ability for oocytes to develop normally and increased vulnerability to abnormalities [72]. This can potentially cause infertility or difficulties during pregnancy.

Moreover, autophagy plays a crucial role in maintaining ovarian function and folliculogenesis, which refers to the recurring process of follicle growth and the release of eggs [36]. The delicate equilibrium between autophagy and apoptosis in granulosa cells, which encircle and provide support to the maturing oocyte within the follicle, is essential for the growth and selection of follicles [28,73]. Dysregulation of autophagy in granulosa cells has been linked to ovarian dysfunction, polycystic ovary syndrome (PCOS), and premature ovarian failure (POF), thus underscoring its importance in reproductive illnesses [74]. Autophagy plays diverse roles in sustaining the health of both the mother and the fetus during pregnancy, thus helping them adapt to constant physiological changes [75]. It promotes the growth and development of the placenta, invasion of trophoblasts, and delivery of nutrients, which are essential for healthy fetal growth and development [76]. In addition, autophagy assists in reducing oxidative stress and inflammation, both of which are elevated during pregnancy [6,77]. This correspondingly provides protection against pregnancy problems such as pre-eclampsia and gestational diabetes.

Furthermore, autophagy plays a role in controlling maternal metabolic changes during pregnancy, including insulin sensitivity, lipid metabolism, and energy balance [78]. Abnormal autophagy has been linked to metabolic abnormalities during pregnancy, thus indicating that it could be a useful target for treating difficulties related to metabolic dysfunction in pregnant women [79]. In addition to its role in reproduction, autophagy also has an impact on hormone release, including effects on gonadotropins such as follicle-stimulating hormone (FSH) and luteinizing hormone (LH), which play crucial roles in regulating ovarian function and the menstrual cycle [80,81]. Imbalanced autophagy in the hypothalamic–pituitary–gonadal (HPG) axis can disturb the equilibrium of hormones, thus resulting in irregular menstrual cycles, infertility, or reproductive problems [81,82]. Moreover, there is growing evidence indicating a connection between autophagy and reproductive aging, which refers to the gradual decrease in fertility and ovarian function as one ages [83]. Decreases in autophagic activity associated with aging can lead to cellular damage, oxidative stress, and genetic instability in the ovary [19,84]. This can ultimately affect fertility and reproductive outcomes in older women. 

## 6. Role of Autophagy in Female Reproductive System Toxicity

Granulosa cells, being the largest and most essential functional cells in the follicle, produce various signaling factors, growth factors, and cytokines [85]. They proliferate, differentiate, and regulate the microenvironment surrounding the follicle, which is crucial for its development and maturation. During follicle development, the number of granulosa cells increases, starting from early sinusoid follicles with only a few thousand cells to antral follicles with a minimum of 100,000 cells [86,87]. Numerous toxicological experiments have been conducted on ovarian granulosa cells to investigate the impact of toxic chemical pollutants such as bisphenol A, acrylamide, pesticides, mycotoxins, zearalenone, BTEX, and microplastics on granulosa cell proliferation and follicle development [15,17,88,89,90,91,92]. Reports showed that chemicals’ toxic effects on the female reproductive system can lead to disorders in the ovarian cycle, increased spontaneous abortion rates, abnormal development of offspring, and a decline in fertility [92,93]. The developmental toxicological mechanisms of these chemicals involve overproduction of reactive oxygen species (ROS), cellular oxidative damage, apoptosis, and autophagy ultimately leading to follicular atresia [17,92,94]. Studies have demonstrated the significance of autophagy in granulosa cells for follicular atresia, as excessive ROS-induced oxidative stress is a major inducer of autophagy in various ovarian cells [17,95]. In fact, toxicants can induce reproductive toxicity through autophagy by increasing the levels of autophagy-related proteins such as Beclin-1, LC3B-II, and the LC3B-II/LC3BeI ratio.

Autophagy and apoptosis play significant roles in hormonally regulated follicular atresia [96]. The former serves as a dual-function mechanism that regulates the survival and death of ovarian cells, resulting in different effects, such as cytoprotection or cytotoxicity [97,98]. Under normal circumstances, autophagy levels in cells are low, promoting cell survival. It acts as an intracellular degradation system that encapsulates unnecessary or malfunctioning cellular components within double-membraned structures called autophagosomes, which are then degraded and recycled [99]. Transmission electron microscopy is the gold standard technique commonly used to observe double-membraned autophagic vacuoles, providing convincing evidence of cell autophagy [17,98]. In TEM images, ovarian cells exposed to toxic chemicals exhibit numerous autophagic vacuoles, that in a later stage fuse with lysosomes for degradation and recycling (Figure 5). Increased autophagy is accompanied by elevated levels of malondialdehyde (MDA) and reduced levels of SOD, GSH, and CAT, indicating that oxidative stress induces autophagy in chemical-exposed ovaries [17,92]. In addition, ROS reportedly activate autophagy in follicular granulosa cells via the mTOR pathway, highlighting the close relationship between oxidative stress and autophagy in ovarian cells [15,16,17,95]. This autophagy’s activation serves as an initial step in maintaining the balance between ROS and antioxidant scavengers, protecting cell viability and playing a cytoprotective role against chemical-induced apoptosis in granulosa cells [100,101]. However, excessive ROS production can result in cell death due to autophagy’s failure to repair ovarian cells. Thus, autophagy can act as an antagonist against apoptosis but can also contribute to cell apoptosis under different conditions [102,103]. This autophagy-induced reproductive toxicity is mediated through the inhibition of the Akt-mTOR signaling pathway, which plays a significant role in autophagic cell death [17,104]. It has also been reported that when autophagy is disrupted in ovarian cells under the effect of toxicants, the transcription factor WT1 accumulated in granulosa cells leads to the inhibition of their differentiation [96].

## 7. Crosstalk between Role of Autophagy on Endometrium and Embryo/Endometrium

Autophagy is a vital cellular process that is responsible for breaking down and reusing cellular components. It has a notable impact on the reproductive system, specifically in the endometrium and its interaction with the embryo [105]. The endometrium, which is the inner lining of the uterus, experiences periodic alterations that are controlled by hormonal signals to prepare for the attachment of an embryo [106]. Autophagy regulates cellular turnover and preserves endometrial homeostasis, creating a favorable environment for the embryo [107]. The interaction between autophagy in the endometrium and the embryo is essential for the effective attachment of the embryo and the occurrence of pregnancy [108]. Autophagy in the endometrium plays a crucial role in mitigating cellular stress, eliminating dysfunctional organelles, and controlling inflammation [109]. These processes are essential for establishing a favorable environment for the embryo. Dysfunction in autophagic mechanisms in the endometrium can result in compromised receptivity, which can contribute to disorders including infertility and recurrent pregnancy loss [110]. Autophagy plays a crucial role in the survival of the embryo during the pre-implantation phase by breaking down cellular components to provide nutrients and energy [111]. The embryonic signals and endometrial autophagy combine to achieve synchrony, facilitating effective implantation [112]. Factors derived from the embryo can influence the process of autophagy in the endometrium, which in turn improves the ability of the endometrium to receive the embryo and facilitates its invasion and attachment [113]. On the other hand, when endometrial autophagy is suitable, it can impact the release of signaling molecules that aid in the growth of an embryo [114]. Gaining insight into the interaction between autophagy in the endometrium and embryo is crucial for the development of therapeutic approaches to tackle reproductive diseases [115]. By specifically focusing on autophagic pathways, it may be feasible to enhance the ability of the endometrium to accept and support embryos, hence improving reproductive outcomes.

## 8. Recent Therapeutic Targets for Autophagy in Female Fertility and Treatment

The process of female fertility is complex and controlled by multiple molecular pathways, one of which is autophagy [58,72]. This process is essential for maintaining the balance and functioning of cells. Recently, there has been a growing emphasis on the understanding of the significance of autophagy in female fertility and its potential as a target for therapeutic intervention in diseases associated with fertility. Recent progress has been made in discovering therapeutic targets of autophagy in female fertility and their implications for treatment options [116]. Due to the significant role of autophagy in female reproductive physiology, a focus on this pathway has become a promising strategy for addressing infertility and associated diseases. Multiple recent studies have demonstrated specific targets within the autophagy pathway that could be utilized for reproductive treatment [53,58]. An example of a target is mTOR, which is a crucial regulator of autophagy and is responsible for controlling cellular development and metabolism. Preclinical studies have demonstrated that suppressing mTOR enhances autophagy and leads to improved egg quality, ovarian function, and reproductive outcomes [117]. AMPK is a cellular energy sensor that controls autophagy based on nutrient availability. AMPK activation has been shown to induce autophagy and enhance oocyte maturation and embryo development [118,119] (Figure 6). Furthermore, it has been suggested that manipulating the expression or function of autophagy-related genes, such as Beclin-1 and Atg7, could be a method to improve fertility in women who are undergoing assisted reproductive technology (ART) [120]. Moreover, recent findings indicate that the disruption of autophagy may play a role in the development of certain fertility problems, such as polycystic ovary syndrome (PCOS), endometriosis, and premature ovarian insufficiency (POI) [121,122]. The targeting of the autophagy mechanisms associated with these diseases could provide new treatment strategies for controlling infertility in affected individuals [123]. Moreover, more investigations are required to clarify the exact mechanisms through which autophagy controls female fertility and to discover other therapeutic targets within this system. The effective translation of these results into clinical practice and the development of autophagy-based medicines for infertility will require collaborative efforts among fundamental scientists, doctors, and pharmaceutical businesses.

### 8.1. Therapeutic Targets for Autophagy Modulation in Female Fertility and Embryo Implantation via Natural Products

The modulation of autophagic activity using drugs or natural substances is a promising strategy for enhancing fertility and optimizing reproductive outcomes. Various substances have been found to be regulators of autophagy, with the ability to either inhibit or stimulate autophagy. Plants, herbs, and traditional medical systems produce natural products that contain a wide range of bioactive chemicals with various pharmacological actions. Several of these substances have demonstrated the ability to regulate autophagy and have potential therapeutic advantages for female fertility (Figure 7). Resveratrol, which is a polyphenolic molecule present in red grapes and berries, has been shown to boost autophagic activity and promote ovarian function in animal models of ovarian aging [124]. Curcumin, which is a beneficial component found in turmeric, has been shown to trigger autophagy and perhaps safeguard against oxidative stress-induced ovarian damage [125]. Furthermore, scientific studies have examined the effects of herbal extracts, including ginseng, green tea, and chasteberry, on the regulation of autophagy and the enhancement of female reproductive health [126,127]. These naturally occurring substances contain phytochemicals that possess antioxidant, anti-inflammatory, and hormone-modulating properties. These features may play a role in their positive impacts on fertility. By specifically focusing on autophagic pathways, these natural substances may provide a safe and efficient option for women who want to improve their fertility and overcome difficulties related to reproduction [128]. Additional investigations are necessary to clarify the mechanisms by which natural compounds impact autophagy and their potential use in reproductive medicine. By utilizing the medicinal properties of organic substances, we can create new methods to enhance women’s reproductive well-being and pregnancy outcomes.

Plants, marine organisms, and microorganisms have a wide range of bioactive chemicals with potential for use in therapy [129]. Various natural substances have been shown to act as regulators of autophagy, thus presenting potential opportunities for improving the process of embryo implantation [130]. Resveratrol, which is a type of polyphenol present in red grapes and berries, has gained significant recognition for its ability to enhance autophagy [131]. Research has shown that it can enhance the growth of embryos and their attachment to the uterus by stimulating autophagic pathways in both embryos and endometrial cells [132]. Curcumin, which is a biologically active molecule extracted from turmeric, has strong anti-inflammatory and antioxidant effects and can regulate autophagy [133]. Curcumin has been demonstrated to enhance autophagic flux, thus leading to improved endometrial receptivity and facilitating embryo implantation in animal models [134]. Moreover, studies have shown that epigallocatechin gallate (EGCG), which is a type of catechin found in green tea, may enhance autophagy and facilitate the process of embryo implantation by exerting its antioxidative and anti-inflammatory effects [135]. Recent research breakthroughs have demonstrated numerous natural substances that can effectively regulate autophagy to improve the process of embryo implantation [132]. Ginsenosides, which are active components extracted from ginseng, have been demonstrated to enhance autophagy and enhance the responsiveness of the endometrium, thereby promoting the successful attachment of embryos [136]. Quercetin, which is a flavonoid found in fruits and vegetables, has a positive impact on implantation by influencing autophagy-related pathways [137]. Natural compounds can enhance embryo implantation by modifying autophagy, and their therapeutic benefits extend beyond this specific property. These chemicals have several effects, thereby affecting different signaling pathways that are involved in the process of implantation and the formation of early pregnancy. For example, polyunsaturated fatty acids, such as omega-3 fatty acids found in fish oil, not only control autophagy but also have anti-inflammatory effects, which help with endometrial receptivity and embryo implantation [138]. To fully utilize the therapeutic benefits of natural products in improving embryo implantation, it is crucial to have a thorough understanding of their molecular mechanisms and pharmacological qualities. By incorporating sophisticated methods such as omics approaches and bioinformatics, we can discover the complex relationships among natural products, autophagy, and implantation. Moreover, it is crucial to conduct thorough preclinical investigations and clinical trials to confirm the effectiveness and safety of therapies based on natural products in enhancing reproductive outcomes. The utilization of the combined benefits of natural products on autophagy and other signaling pathways has great potential for enhancing fertility therapies and reducing difficulties related to implantation failure.

### 8.2. Targeting Autophagy Modulation in Female Fertility and Embryo Implantation via Synthetic Compounds

Synthetic substances that specifically affect the process of autophagy show great potential for use in medical treatments to improve female fertility and enhance the success of embryo implantation (Figure 7). Preclinical studies have demonstrated that small compounds, such as rapamycin and spermidine, can regulate autophagic activity and enhance reproductive outcomes [139]. These drugs exert their effects by activating several signaling pathways, including the mTOR and AMP pathways, which play crucial roles in regulating autophagy. Furthermore, the development of new artificial substances that particularly target autophagy in the female reproductive system has significant potential for clinical application [140]. By specifically controlling the flow of autophagy in oocytes, granulosa cells, and endometrial cells, these substances have the potential to alleviate reproductive problems and improve fertility. In addition, the use of delivery vehicles, such as nanoparticles and liposomes, could enhance the effectiveness and selectivity of autophagy-modulating drugs [141]. This scenario would reduce unintended effects and maximize therapeutic advantages. Nevertheless, there are still obstacles to overcome regarding the application of autophagy regulation in clinical settings for the purpose of addressing female infertility and facilitating embryo implantation. Additional investigations are required to clarify the exact mechanisms that cause autophagy dysregulation in reproductive diseases and to discover new targets for treatments. Furthermore, comprehensive preclinical and clinical investigations are important for evaluating the safety, effectiveness, and enduring impacts of synthetic drugs that specifically target autophagy in the female reproductive system [142]. By optimizing autophagic activity in the oocyte and the endometrium, these chemicals provide a new and effective treatment method for reproductive diseases and may improve assisted reproductive procedures [143]. Continued research endeavors in this domain are crucial for fully harnessing the potential of autophagy modulation in female reproductive health.

### 8.3. Potential Diagnostic and Therapeutic Applications of Autophagy in Female Fertility

Monitoring autophagy indicators can offer vital information about the condition of the ovaries and their ability to reproduce [4]. Autophagy-related genes (ATGs) and proteins, including LC3, Beclin-1, and p62, play a vital role in the quality of oocytes and the development of follicles [72,144]. Anomalous expression of these markers can suggest hindered egg maturation or diminished ovarian reserve, assisting in the identification of diseases such as PCOS and POF [145]. Moreover, assessing autophagic activity in granulosa cells, which provide support for oocyte development, can function as a prognostic indicator for the achievement of fertilization and the implantation of embryos in ART [146].

Targeting the mechanisms involved in autophagy has potential for improving female fertility. Regulating autophagy can enhance the quality of oocytes and their ability to mature, which is essential for achieving optimal pregnancy results [147]. Autophagy inducers, like resveratrol and spermidine, have demonstrated promise in stimulating oocyte maturation and safeguarding against oxidative stress [148]. On the other hand, inhibiting autophagy could be advantageous in situations such as endometriosis, where excessive autophagy promotes the survival of cells in abnormal endometrial tissues [149]. Moreover, the manipulation of autophagy can aid in maintaining ovarian function while undergoing cancer treatments [50]. Chemotherapy and radiotherapy frequently result in ovarian impairment and infertility [150]. Autophagy activators can alleviate these consequences by safeguarding the ovarian reserve and bolstering tissue repair processes [151]. The dual function of autophagy in both facilitating and impeding reproductive processes highlights its potential as a diagnostic and therapeutic target in female fertility [152]. Further investigation is crucial to completely understand the mechanisms involved in autophagy in reproductive biology, which would facilitate the development of new therapies that could enhance fertility outcomes and promote ovarian health in women.

## 9. Limitations and Future Perspectives on Autophagy in Female Fertility

The connection between autophagy and female fertility has received considerable attention in recent years. Autophagy plays a role in multiple reproductive processes, such as egg development, folliculogenesis, and embryo implantation [153]. Nevertheless, despite the progress made in comprehending this subject, there are still several constraints in elucidating the complex interaction between autophagy and female fertility. The constraints and deliberations on potential prospects and limitations for advancing research on the role of autophagy in female fertility are subsequently discussed.

Insufficient comprehension of autophagic mechanisms in reproductive tissues is a substantial obstacle due to the intricate control of autophagy. Although the importance of autophagy in preserving oocyte quality and controlling follicular growth has been acknowledged, the exact mechanisms underlying these processes are not fully understood. Further research is needed to understand the specific molecular processes that control autophagy in relation to female fertility. The investigation of autophagy in reproductive tissues presents significant technological obstacles. Existing techniques for evaluating autophagic activity, such as immunoblotting and immunofluorescence, suffer from a lack of specificity and can produce inconsistent findings. It is crucial to develop more precise and reliable methods that are specifically designed for evaluating autophagy in reproductive tissues. Although preclinical studies have indicated a connection between the disruption of autophagy and female infertility, there is a lack of clinical data to substantiate this correlation. There is a need for extensive clinical investigations to examine biomarkers related to autophagy in individuals who are unable to conceive to confirm the significance of autophagy in human fertility and discover possible targets for diagnosis and treatment. The impact of the reduction in autophagic activity associated with aging has been linked to reproductive aging and age-related infertility [154]. Nevertheless, the specific consequences of a decrease in autophagy on the quality of oocytes and the results of reproduction are not yet fully understood. To understand the age-related alterations in autophagy and their consequences for female fertility, it is necessary to conduct longitudinal research involving animal models and human cohorts.

To overcome this limitation, advanced methods in molecular biology are used to evaluate autophagy. Novel molecular methodologies, such as single-cell RNA sequencing and proteomics, have the potential to decipher the intricate regulatory networks of autophagy in reproductive organs. By combining high-throughput methods with functional assays, we can gain a thorough understanding of the dynamics of autophagy during oogenesis, folliculogenesis, and embryogenesis [154]. Advances in targeted autophagy modulators, which specifically target certain elements of the autophagic machinery, exhibit considerable potential as therapeutic approaches for addressing female infertility. The development of small compounds or biologics that selectively stimulate or suppress autophagy in certain tissues can boost the quality of oocytes, improve ovarian function, and optimize reproductive results [155]. The utilization of CRISPR/Cas9-mediated genome editing has exceptional potential for exploring the functional importance of autophagy-related genes in female fertility [156]. Genetically modified animal models with precise modifications in autophagy-related pathways will allow researchers to analyze the specific functions of autophagy in reproductive processes [78,157]. Furthermore, the performance of longitudinal studies that monitor autophagic activity over the entire lifespan of females is crucial for comprehending the loss in fertility that occurs with age. By integrating autophagy assessments with clinical outcomes in various groups, we can clarify the causal connection between autophagy malfunction and reproductive aging.

## 10. Conclusions

To understand female fertility, this review explored autophagy and its enormous effects on reproductive health. Our study provides important insights into the intricacy and potential of autophagy and female fertility. Dynamic autophagy modulation during reproductive stages is a key discovery. Autophagy maintains cellular homeostasis throughout follicular growth, fertilization, and implantation, thus allowing for adaptation to the reproductive microenvironment [158]. This multifaceted overview of autophagy challenges its static image and highlights its responsiveness to female fertility. The elucidation of this link suggests new paths for therapies to improve oocyte quality, which is a key component in age-related decreases in fertility and recurrent pregnancy loss [10]. An understanding of the molecular mechanisms of autophagy and regulatory pathways involved in female fertility provide opportunities for new treatments. Those with reproductive issues may find hope in the targeting of autophagic systems to reduce age-related fertility. This extensive research provides the groundwork for reproductive health-tailored treatments. The tailoring of reproductive therapies to the individual’s autophagic profile can enhance success rates and reduce dangers and negative effects. Its potential applications extend beyond assisted reproductive technology, which can affect fertility-related diseases. However, the complex relationship between autophagy and female fertility provides a promising scientific and clinical picture. The dynamic nature of autophagy, its impact on oocyte quality, and its translational implications necessitate further investigations. At the intersection of basic science and practical innovation, the elucidation of autophagy in female fertility has encouraged researchers, clinicians, and policymakers to work together to develop customized, effective reproductive treatments. We must acknowledge the gaps in our understanding of autophagy in female reproductive health as we begin this exploration. Future research will encounter obstacles and possibilities in understanding the role of autophagy in reproduction. By identifying these gaps, we hope to motivate future research into the function of autophagy in reproductive biology and provide novel treatment possibilities.

## Figures and Tables

**Figure 1 cells-13-01354-f001:**
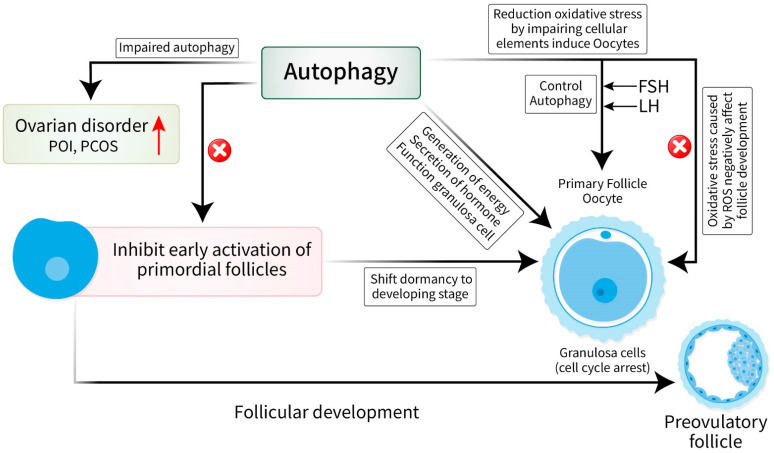
Relationship between autophagy and follicular development. Autophagy has a significant impact on follicular development ranging from primordial to preovulatory stages. Autophagy inhibits early activation of primordial follicles thus controlling the quality and quantity of it. It acts on granulosa cells resulting in the generation of energy and secretion of hormones controlled by gonadotropins. Moreover, autophagy inhibits ROS, which is produced by follicles and negatively affects the quality of oocytes. On the other hand, impaired autophagy may result in POI and PCOS.

**Figure 2 cells-13-01354-f002:**
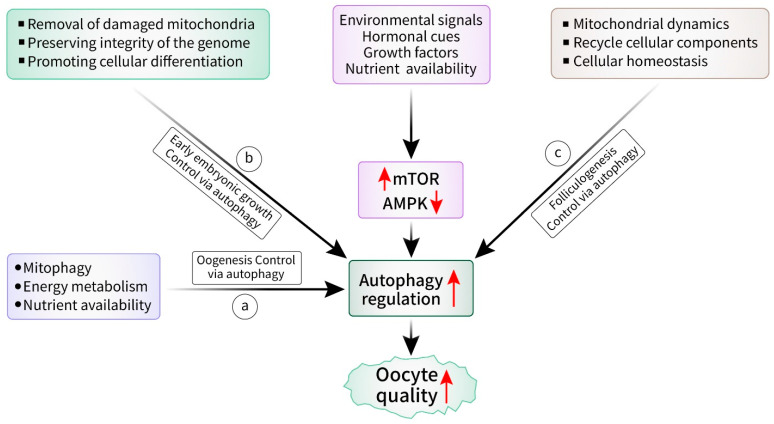
Autophagy control in oocyte quality via oogenesis, early embryonic growth and folliculogenesis. Specific environmental, external, and internal factors activate mTOR and AMPK pathways that result in the regulation and control of autophagy. Autophagy in oocytes is broadly involved in three interconnected phases: (a) In oogenesis: Malfunctioning mitochondria is eliminated, a process known as mitophagy, to avoid oxidative stress and impaired oocyte quality. In addition, autophagy provides adequate levels of ATP via metabolic substrates. Furthermore, it breaks down and recycles cellular components for nutritional requirements. (b) In early embryonic growth: Autophagy removes parental mitochondria since the mixture of parental mitochondria can lead to mitochondrial dysfunction. On the other hand, it preserves the integrity of the genome through removing damaged DNA, clearing oxidative stress, and regulating the repair of DNA. It also promotes cellular differentiation. (c) In folliculogenesis: Autophagy maintains cellular homeostasis by breaking down and recycling cellular components or eliminating damaged organelles.

**Figure 3 cells-13-01354-f003:**
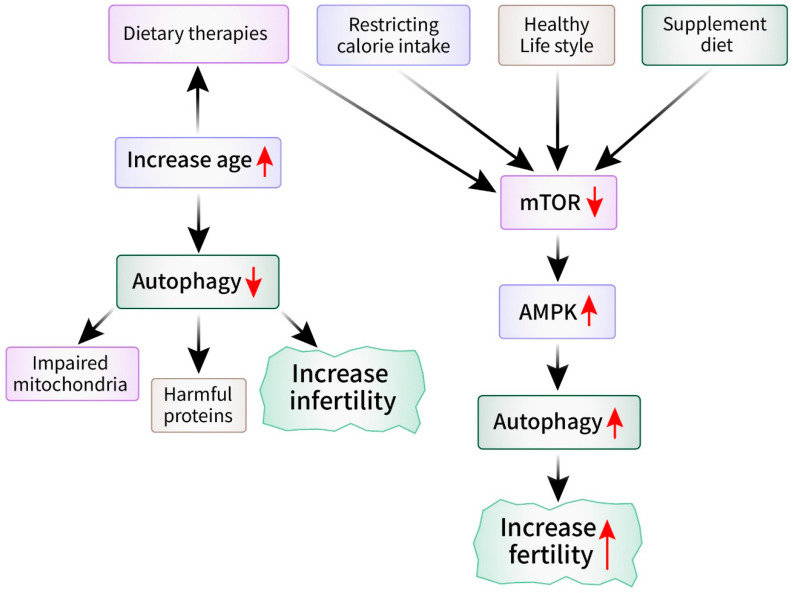
Role of autophagy in reproductive aging. Several internal and external factors upregulate autophagy by decreasing mTOR and increasing AMPK. Increased autophagy maintains cellular homeostasis that consequently inhibits reproductive aging. In contrast, impaired autophagy has been linked to lower oocyte and sperm quality, infertility, impaired cell structure, and production of harmful proteins that accelerate reproductive aging.

**Figure 4 cells-13-01354-f004:**
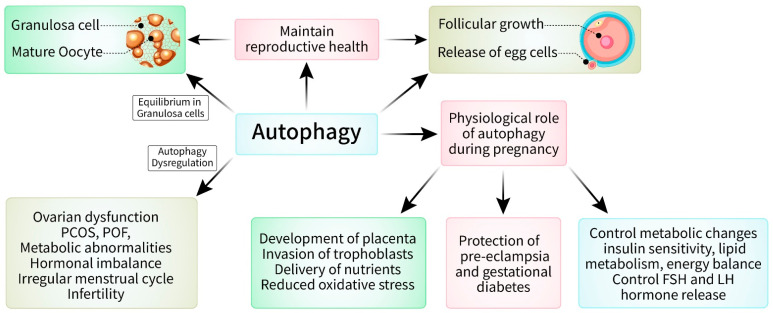
Physiological role of autophagy in female reproductive system. Autophagy has been linked to playing a crucial role in four areas: fertility, reproductive health, pregnancy, and metabolic changes. In fertility: autophagy helps in growth and maintenance of oocytes where impaired autophagy may result in abnormal development of oocytes. In reproductive health: autophagy keeps equilibrium in granulosa cell and impaired one causes ovarian distinction, PCOS, POF, cellular damage and oxidative stress. In pregnancy: autophagy develops placenta, helps in invasion of trophoblasts, etc., where impaired autophagy is associated with preeclampsia and gestational diabetes. In metabolic changes: normal autophagy regulates insulin sensitivity, lipid metabolism, and energy balance. Infraction of autophagy leads to irregular menstrual cycle.

**Figure 5 cells-13-01354-f005:**
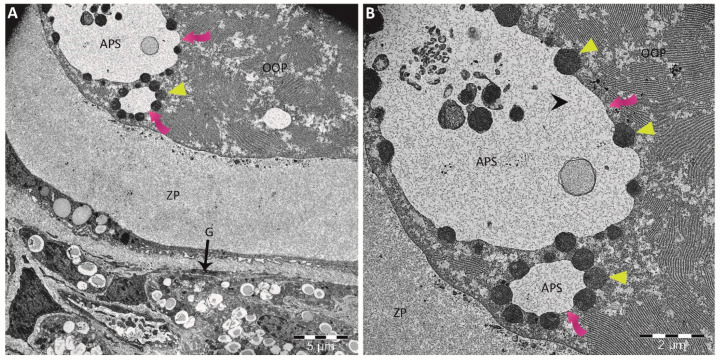
Transmission electron microscopy (TEM) images providing direct visual evidence of the autophagic process occurring within the oocyte cytoplasm. (**A**) The ultrastructure of an ovary from a rat shows the presence of autophagosomes (pink arrows) within the cytoplasm of an oocyte, which appears to be a protective mechanism after in utero treatment with acrylamide. (**B**) A higher-magnification microphotograph of the image in Figure (**A**) reveals large autophagosomes (APS) engulfing extracellular material. These autophagosomes are surrounded by many lysosomes (yellow arrowheads) at different stages of fusion, forming autophagolysosomes. APS: autophagosomes; G: granulosa cells; OOP: ooplasm; ZP: zona pellucida.

**Figure 6 cells-13-01354-f006:**
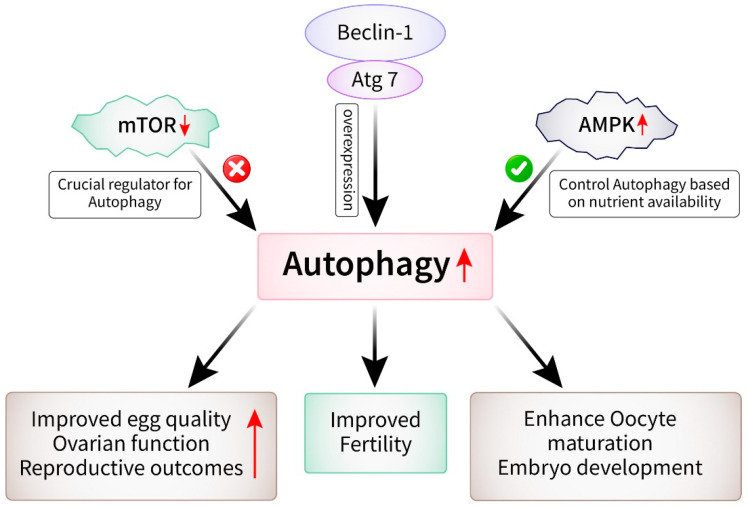
Recent therapeutic targets for autophagy in female fertility and treatment. Genes and proteins: Beclin-1 protein and Atg7 gene have an impact on autophagy that leads to improved fertility in those undergoing ART. Pathways: downregulated mTOR increases autophagy that results in improved egg quality, ovarian function, and reproductive outcomes. On the other hand, the upregulation of AMPK is associated with oocyte maturation and embryo development.

**Figure 7 cells-13-01354-f007:**
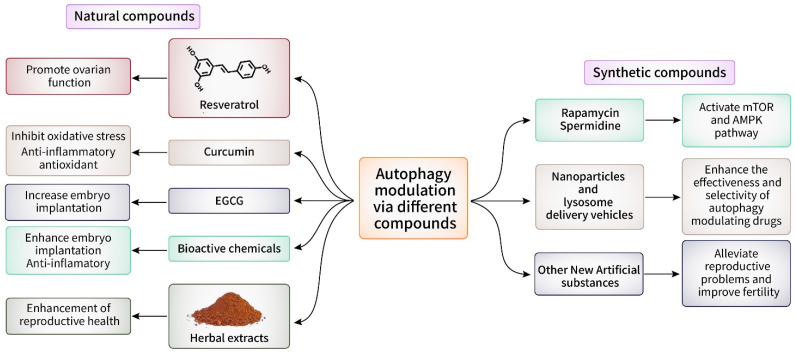
Autophagy modulation via natural and synthetic compounds. Both natural and synthetic compounds play a significant role in autophagy modulation. Natural compounds like resveratrol, curcumin, EGCG, bioactive chemicals and herbal extracts have numerous positive impacts on autophagy modulation including promotion of ovarian function, inhibition of oxidative stress, and enhancement of female reproductive health. They also possess anti-inflammatory, antioxidant, and hormone-modulating properties. One the other hand, synthetic compounds like rapamycin, spermidine, and other new artificial substances with their carriers (nanoparticles and lysosome) activate mTOR and AMPK pathways, lessen reproductive problems, and improve fertility by enhancing the effectiveness and selectivity of autophagy-modulating drugs.

## Data Availability

The datasets supporting the conclusions of this study are included within the article.

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
