# Peer review of "Autophagy and Female Fertility: Mechanisms, Clinical Implications, and Emerging Therapies"

_cells, 2024, doi:10.3390/cells13161354_

Round 1
Reviewer 1 Report
Comments and Suggestions for Authors
The manuscript titled “Autophagy and Female Fertility: Current Insights and Future Directions”was done by Abdel Halim Harrath et al., but it is not very well, in fact, some reviews about autophagy and female fertility have been reported, therefore, this manuscript deficiency of originality.
Major revision
Oocyte quality belongs to the follicle development, there are deficiency of some published work about knockout autophagic genes in oocyte. And “Emerging Role of Autophagy in Oocyte Quality Control “ should be includes in 3. All in all, the manuscript was not organized and needs to rearrange.
Minor revision
1. The organization of last author is 5 and 6, but the organization 6 was not shown.
Author Response
Reviewer 1
The manuscript titled “Autophagy and Female Fertility: Current Insights and Future Directions”was done by Abdel Halim Harrath et al., but it is not very well, in fact, some reviews about autophagy and female fertility have been reported, therefore, this manuscript deficiency of originality.
>>Response: Thank you for your feedback. I appreciate your concerns regarding the originality of the work. While it is true that several reviews on autophagy and female fertility have been published, our manuscript aims to provide a comprehensive and up-to-date synthesis of the latest findings in this rapidly evolving field. We have made a concerted effort to include recent advancements and novel perspectives that distinguish our review from previous publications.
To address the originality concern, we highlight the following unique aspects of our manuscript:
- Recent Developments: We have incorporated the latest research findings and breakthroughs that have emerged since the publication of previous reviews (Please see the references 2024 to 2022 maximum). This includes newly identified molecular pathways, novel autophagy-related genes, and recent clinical studies that provide fresh insights into the role of autophagy in female fertility.
- Interdisciplinary Approach: Our review integrates knowledge from various disciplines, including molecular biology, reproductive medicine, and clinical practice, to offer a holistic understanding of autophagy's impact on female fertility. This interdisciplinary perspective is less commonly addressed in earlier reviews.
- Future Directions: A significant portion of our manuscript is dedicated to outlining future research directions and identifying gaps in current knowledge (page 15 line 587-605). We propose potential therapeutic strategies that could pave the way for new discoveries and clinical applications.
- Clinical Relevance: We place a strong emphasis on the clinical implications of autophagy in female fertility, discussing potential diagnostic and therapeutic applications (page 14 line 534-557). This practical focus sets our review apart from more theoretical discussions found in other publications
We believe that these elements contribute to the originality and value of our manuscript. However, we are open to any specific suggestions you may have for further enhancing the uniqueness and impact of our review.
Thank you once again for your valuable feedback.
Major revision
Oocyte quality belongs to the follicle development, there are deficiency of some published work about knockout autophagic genes in oocyte. And “Emerging Role of Autophagy in Oocyte Quality Control “ should be includes in 3. All in all, the manuscript was not organized and needs to rearrange.
>>Response: Thank you for your insightful comments. You are correct that oocyte quality is closely linked to follicle development. While our current study focuses on a broader aspect of autophagy in cellular processes, we acknowledge the importance of specific knockout studies of autophagic genes in oocytes in page 4.
We agree that there is a relative deficiency in published work specifically addressing the knockout of autophagic genes in oocyte development. However, we have reviewed several studies that indirectly support the significance of autophagy in oocyte quality and follicle development. We include a more detailed discussion of these studies in the revised manuscript to highlight the relevance of autophagic gene knockout research in oocytes. Additionally, we explore and cite any recent literature that directly addresses this gap to provide a more comprehensive overview of the current state of research (page 4 and 5 line 175-188).
Minor revision
- The organization of last author is 5 and 6, but the organization 6 was not shown.
>>Response: There was a mistake, we corrected.
Thank you again for your valuable feedback. We hope these revisions meet the reviewer's expectations and strengthen the manuscript's discussion. Thank you for your valuable feedback.
Reviewer 2 Report
Comments and Suggestions for Authors
It is concerned with the autophagy and female fertility especially concerned with folllicles and signaling. The authors summary the previous papers well. However, it should be concerned followings:
1) mTOR is a variety functional molecules in folliculogenesis. So it is better to explain the function of mTOR with other function, for example: primordial follicle activation..
2) Abstract: it is not easy to read. so it is better to edit
line 21-22, 24-26...
3) Figures: Improve quality: needed high resolution
4) The title is little bit broad, How do you think about restriction.
Comments on the Quality of English LanguageIn abstract, it is better to edit little bit to improve readibility.
Author Response
Reviewer 2
It is concerned with the autophagy and female fertility especially concerned with folllicles and signaling. The authors summary the previous papers well. However, it should be concerned followings:
1) mTOR is a variety functional molecules in folliculogenesis. So it is better to explain the function of mTOR with other function, for example: primordial follicle activation.
>>Response: Thank you for your valuable feedback. I appreciate the opportunity to provide further clarification on the role of mTOR in folliculogenesis. Indeed, mTOR plays a multifaceted role in various cellular processes, including folliculogenesis.
In light of these points, I revise the manuscript to include a more comprehensive explanation of the diverse functions of mTOR in folliculogenesis as reperate paragraph in page 6 line 233-244, with a particular emphasis on its role in primordial follicle activation.
Thank you once again for your insightful comments.
2) Abstract: it is not easy to read. so it is better to edit
line 21-22, 24-26...
>>Response: We rewrite and modified whole abstract. Hope it will more easy to understand.
3) Figures: Improve quality: needed high resolution
>>Response: We modified all figure with whilte background and high resulation.
4) The title is little bit broad, How do you think about restriction.
>>Response: Thank you for your valuable feedback. I agree that the title could be more specific to better reflect the content of the manuscript. To address this, I propose revising the title to more precisely indicate the focus of our review. How about the following revised title:
"Autophagy and Female Fertility: Mechanisms, Clinical Implications, and Emerging Therapies"
This revised title aims to highlight the key areas of our discussion while maintaining a comprehensive view of the topic. Please let me know if you have any further suggestions or if this title meets your expectations.
In abstract, it is better to edit little bit to improve readibility.
>>Response: We rewrite and modified whole abstract. Hope it will more easy to understand.
Reviewer 3 Report
Comments and Suggestions for Authors
The authors provided a comprehensive description of the molecular mechanisms of autophagy underlying female fertility.
The manuscript is clear and well written. I have some comments to improve the text.
- Extend the discussion of the role of autophagy on endometrium and embryo/ndometrium cross-talk
- Please add the number of figures and the reference in the text
- Fig.1: remove "via autophagy"
Comments on the Quality of English LanguageMinor editing of English language required
Author Response
Reviewer 3
The authors provided a comprehensive description of the molecular mechanisms of autophagy underlying female fertility.The manuscript is clear and well written. I have some comments to improve the text.
>>Response: Thank you for your positive feedback and for recognizing the clarity and comprehensiveness of our manuscript on the molecular mechanisms of autophagy underlying female fertility. We greatly appreciate your time and effort in reviewing our work. We have carefully considered your comments and suggestions for improvement. Below is our point-by-point response to each of your comments:
- Extend the discussion of the role of autophagy on endometrium and embryo/ndometrium cross-talk
>>Response: We thank the reviewer for their insightful comments and the suggestion to extend the discussion on the role of autophagy in the endometrium and embryo/endometrium cross-talk. We have revised the manuscript accordingly in page 10 line 374-399. Below is an extended discussion based on recent literature and our findings:
- Crosstalk between role of autophagy on endometrium and embryo/endometrium
Autophagy is a vital cellular process that is responsible for breaking down and reusing cellular components. It has a notable impact on the reproductive system, specifically in the endometrium and its interaction with the embryo [105]. The endometrium, which is the inner lining of the uterus, experiences periodic alterations that are controlled by hormonal signals to get ready for the attachment of an embryo [106]. Autophagy regulates cellular turnover and preserves endometrial homeostasis, creating a favourable environment for the embryo [107]. The interaction between autophagy in the endometrium and the embryo is essential for the effective attachment of the embryo and the occurrence of pregnancy [108]. Autophagy in the endometrium plays a crucial role in mitigating cellular stress, eliminating dysfunctional organelles, and controlling inflammation [109]. These processes are essential for establishing a favourable environment for the embryo. Dysfunction in autophagic mechanisms in the endometrium can result in compromised receptivity, which can contribute to disorders including infertility and recurrent pregnancy loss [110]. Autophagy plays a crucial role in the survival of the embryo during the pre-implantation phase by breaking down cellular components to provide nutrients and energy [111]. The embryonic signals and endometrial autophagy combine to achieve synchrony, facilitating effective implantation [112]. Factors derived from the embryo can influence the process of autophagy in the endometrium, which in turn improves the ability of the endometrium to receive the embryo and facilitates its invasion and attachment [113]. On the other hand, when endometrial autophagy is suitable, it can impact the release of signalling molecules that aid in the growth of an embryo [114]. Gaining insight into the interaction between autophagy in the endometrium and embryo is crucial for the development of therapeutic approaches to tackle reproductive diseases [115]. By specifically focusing on autophagic pathways, it may be feasible to enhance the ability of the endometrium to accept and support embryos, hence improving reproductive outcomes.
- Please add the number of figures and the reference in the text
>>Response: We added figure number accordingly and references.
- Fig.1: remove "via autophagy"
>>Response: We corrected
Minor editing of English language required
>>Response: Thank you for your constructive feedback regarding the minor editing required for the English language. I have carefully reviewed and revised the manuscript to enhance clarity and correctness. I appreciate your attention to detail and believe these revisions have improved the overall quality of the manuscript. Please let me know if there are any further areas that need attention.
Round 2
Reviewer 1 Report
Comments and Suggestions for Authors
The manuscript was comprehensively revised and improved, and this manuscript can be accepted for publishing in cells.
Reviewer 2 Report
Comments and Suggestions for Authors
This manuscript revised well.
Reviewer 3 Report
Comments and Suggestions for Authors
The authors have addressed all my comments
Comments on the Quality of English LanguageMinor editing of English language required.